# Epidemiology of Infectious Bursal Disease Virus in Poland during 2016–2022

**DOI:** 10.3390/v15020289

**Published:** 2023-01-19

**Authors:** Anna Pikuła, Anna Lisowska, Katarzyna Domańska-Blicharz

**Affiliations:** Department of Poultry Diseases, National Veterinary Research Institute, al. Partyzantów 57, 24-100 Pulawy, Poland

**Keywords:** infectious bursal disease virus, molecular survey, genetic evolution, reassortment

## Abstract

Infectious bursal disease virus is an immunosuppressive ubiquitous pathogen that causes serious economic losses in poultry production. The virus is prone to genetic changes through mutations and reassortment, which drive the emergence of new variants and lead to a change in the epidemiological situation in a field. Such a situation is currently being reported due to a large wave of IBDV A3B1 reassortant infections in northwestern Europe. On the other hand, in Poland, which is the largest producer of chicken meat in the EU, the IBDVs of genotypes A3B2 and A3B4 were circulating just before the emergence of A3B1 reassortants. The purpose of the presented study was to update the IBDV epidemiological situation. The performed molecular survey based on the sequence of both genome segments showed the presence of very virulent strains (A3B2) and reassortants of genotypes A3B4 and A3B1; moreover, two of these genotypes are newly introduced IBDV lineages. In addition, a number of amino acid substitutions were demonstrated, including within antigenic epitopes and virulence determinants. In conclusion, the results obtained indicated a dynamic epidemiological situation in Poland, which highlights the need for further monitoring studies in the region and verification of protection conferred by the vaccines used against infection with detected IBDV.

## 1. Introduction

The infectious bursal disease virus (IBDV) is a ubiquitous pathogen that leads to economic losses in poultry production worldwide. The virus replicates in immature B cells in the bursa of Fabricius, causing immunosuppression [1]. IBDV infection induces the so-called Gumboro disease (IBD), and the severity of the course of the disease (subclinical/acute) depends on the pathotype of the virus, the immune status, age, and the breed of the chickens [2].

IBDV is a representant of the *Avibirnavirus* genus within the *Birnaviridae* family. The virus genome is built of a double-stranded RNA divided into two segments A and B, which encodes five proteins [3]. The polycistronic segment A encodes the non-structural VP5 protein and the VP2-VP4-VP3 polyprotein, which is post-translationally cleaved by the viral protease VP4 [4]. The VP2 is the capsid protein, which contains antigenic [5] and virulence [6] determinants located within the P domain (aa 206–350). Segment B encodes viral polymerase (RdRp, VP1) responsible for replication, and transcription [7], together with the VP3 protein translation [8]; however, it also contributes to viral virulence [9]. In addition, this non-enveloped virus is highly resistant to environmental conditions, which facilitates its persistence in poultry farms and further spread.

IBDVs show genetic, pathotypic, and antigenic diversity; thus, continuous monitoring of circulating filed IBDV strains is epidemiologically important. Recently, rules have been established for IBDV genotyping based on the partial sequence of both segments [10,11], resulting in the detection of 15 IBDV genotypes. The prophylactic vaccinations of chicken flocks and maintenance of biosecurity measures are the primary tools for disease control. However, widely applied immunoprophylaxis does not effectively prevent IBDV infection, since the virus remains detected worldwide. Furthermore, the vulnerability of the virus to frequent mutations and reassortment enhanced by vaccine selection pressures contributes to virus evolution, as evidenced by the occurrence of different genotypes [12,13]. Moreover, changes in the epidemiological landscape of IBDV have been frequently observed. In China, for example, the novel A2dB1 variant strains and reassortants of the A3B3 genotype have now been detected [14]; furthermore, the variant strains have spread further into Japan and South Korea [15,16,17]. In contrast, the emergence and spread of the novel A3B1 reassortants have been reported across western Europe [18,19], while the very virulent IBDV (vvIBDV) continues to dominate in Africa [20,21], and the reassortants of the A3B5 genotype have been found in Nigeria [22]. These examples illustrate the need for the continuous monitoring of poultry flocks for IBDV infection in order to improve the effectiveness of control measures.

The aim of this study was to update the IBDV epidemiological situation in Poland in the context of the wave of infection with novel A3B1 reassortants in Europe. Furthermore, the field IBDV strains were molecularly characterized.

## 2. Materials and Methods

### 2.1. Samples

A total of 1359 samples were collected from 216 broiler and laying chicken flocks raised in 13 administrative areas of Poland for laboratory investigation between January 2016 and March 2022. The samples were sent to NVRI for diagnostic testing and were taken from flocks suspected of being infected with IBDV, MDV, or adenovirus. The bursal or spleen tissue obtained from one flock was pooled (from 2 to 12), homogenized in phosphate-buffered saline (4 mL PBS per gram tissue), and centrifuged at 4 °C 3500× *g* for 15 min. For the preparation of bursal tissue imprints on FTA cards, 5 mm diameter discs were cut and placed in a sterile tube and incubated with 1 mL of PBS for 1 h at room temperature. All post-treatment samples were stored below −65 °C for further examination.

### 2.2. Detection and Identification of IBDV

Viral RNA was extracted using an IndiSpin Pathogen Kit (Indical Bioscience, Germany) following the supplier’s instructions. The presence of IBDV was confirmed using the previously published real-time RT-PCR protocol [23]. The initial differentiation into vaccine and field IBDV strains was based on the partial amplification of the VP2 gene [24], followed by Sanger sequencing and sequence comparison to the vaccine strains. For the genotyping of field strains, the partial amplification of the VP1 gene was performed using the primers VP1.154F and VP1.998R. RT-PCR reaction was carried out using a one-step RT-PCR kit (Qiagen, Germany), according to the manufacturer’s guidelines and the following thermal conditions: RT at 50 °C for 30 min, PCR—initial denaturation at 95 °C for 15 min, followed by 35 cycles of amplification (denaturation 94 °C for 40 s, annealing at 57 °C for 1 min, and elongation at 72 °C for 1 min). The final extension was conducted at 72 °C for 7 min. Furthermore, the full-coding sequences of both genome segments (primers SegA_F/SegA0_R and SegB0_F/SegB0_R) of the selected strains were obtained using a SuperScript IV One-Step RT-PCR System with Platinum SuperFi DNA Polymerase (Invitrogen, Vilnius, Lithuania), according to the manufacturer’s instructions. The obtained amplicons after purification with QIAquick Gel Extraction Kit (Qiagen, Hilden, Germany) were cloned into the pJET1.2 vector (Thermo Scientific, Vilnius, Lithuania). The sequences of all the primers used in the study are listed in Table 1. All the obtained PCR products and recombinant plasmids were sequenced in both directions using a commercial service (Genomed, Warsaw, Poland).

### 2.3. Phylogenetic Analysis

The partial sequences of the VP2 and VP1 genes or the full coding sequences of the A and B segments of 180 IBDV reference strains representing different genotypes were downloaded from the GenBank database and included in the analyses (Appendix A). The ClustalW algorithm implemented in MEGAX software was used to align all sets of the sequence [25]. IQ-TREE software (version 1.6.12) was used both to estimate the best evolutionary model and to infer phylogenetic trees using the maximum likelihood algorithm, and confidence levels for the branches were determined with the Shimodaira–Hasegawa test and 1000 replicates of bootstrap [26]. The nucleotide identity was calculated using Geneious Prime 2022.1.1, while tree visualization was performed using the iTOL v6 online tool [27].

### 2.4. Virus Isolation

The virus isolation was performed using 10-day-old specific pathogen-free (SPF) embryonated eggs (Valo BioMedia, Osterholz-Scharmbeck, Germany) and the chorioallantoic membrane (CAM) route. Before inoculation, the tissue supernatant was filtered through a 0.45 µm membrane. After a 7-day incubation, the embryos and chorioallantoic membrane were harvested, homogenized, and stored below −65 °C.

## 3. Results

### 3.1. Molecular Detection, Genotyping, and Characterization of IBDV

The molecular survey revealed the presence of IBDV in 196 flocks, among which field and vaccine strains were detected in 52 and 144 flocks, respectively. The number of flocks with the presence of field IBDV accounted for 24.07% of the total surveyed (52/216). By year, the percentage of positive flocks was 12.8% (6/47) in 2016, 30.3% (20/60) in 2017, 33.3% (16/48) in 2018, 9.5% (2/20) in 2019, 29.4% (5/17) in 2020, 6.7% (1/14) in 2021, and 100% in 2022 (2/2). The detailed data on flocks where such strains were found are presented in Table 2. Genotyping analysis based on the partial sequence of both segments (Figure 1) showed that the detected Polish IBDV strains represent the A3B2 genotype, i.e., vvIBDV, and the reassortants of the A3B4 and A3B1 genotypes. The administrative location of the demonstrated genotypes is shown in Figure 2. In addition, in one case (strain H286/17), a serotype 2 strain was confirmed (Figure 1). The obtained sequences were deposited in the GenBank database (accession numbers presented in Table 2). The reassortants of the A3B4 genotype were the most abundant, and the survey conducted confirmed its presence in 27 flocks throughout the sampling period (*n* = 6 in 2016, *n* = 9 in 2017, *n* = 8 in 2018, *n* = 3 in 2020, and *n* = 1 in 2022). A total of 20 chicken flocks infected with very virulent IBDV (A3B2 genotype) were identified between 2017 and 2020 (Table 1, *n* = 9 in 2017, *n* = 8 in 2018, *n* = 1 in 2019, and *n* = 2 in 2020). In turn, two reassortant strains of the A3B1 genotype were confirmed in 2021 and 2022 in broiler flocks reared in western Poland (Figure 2, Kuyavian–Pomeranian and Greater Poland Voivodeships). For the three strains 97/1967/17, H286/17, and 68/19, the VP1 sequence was not obtained, and therefore genotyping is incomplete.

The phylogenetic analysis based on partial VP2 nucleotide sequences showed that the Polish strains clustered in three separate branches within genogroup A3, which were designated as A3.1, A3.2, and A3.3 (Figure 3), while for VP1, they clustered in three different genogroups: B1, B2, and B4 (Figure 1B). Cluster A3.1 grouped the reassortants of the A3B1 genotype, and cluster A3.2 gathered the very virulent A3B2 strains, while the collected strains in cluster A3.3 represented the A3B4 genotype, indicating separate evolution of the VP2 gene of these three Polish IBDV lineages despite belonging to the same genogroup A3.

The two Polish IBDV reassortants of genotype A3B1 showed relatively low similarity to each other, 97.1% and 98.6% for VP2 and VP1, respectively (Appendix A). This very likely points to two independent introductions of this virus in Poland. The 18/21/Poland/2021 strain had the highest nucleotide similarity with A3B1 reassortants reported by Mato et al. [19] circulating in Sweden, Germany, Belgium, and the Czech Republic in 2017–2019 (99.3–99.5% for partial VP2; the VP1 sequences analyzed had short overlapping regions; hence, their similarity was not determined; Appendix A), while strain 38/22/Poland/2022 was more similar to the German and Dutch strains from the survey conducted in 2021 by Legnardi et al. [18] (99.3–99.5% and 99.7% for VP2 and VP1, respectively; Appendix A).

The clade A3.2 was formed by the strains of genotype A3B2 (Figure 3); however, phylogenetic analysis showed that the very virulent strains both described in this paper and circulating in 2014–2015 were separately clustered, which may indicate subsequent introductions of vvIBDV (Figure 1A). This hypothesis is supported by the high homogeneity of this group of viruses at the nucleotide level, where the similarity for VP2 and VP1 genes was 98.4–100% and 99.1–100%, respectively. Whereas the identity for VP2 with other polish A3 strains was much lower and ranged from 94.3% to 96.4% for A3.3 and from 94.1% to 95.2% for A3.1. The similarity to the 2014–2015 vvIBDV strains was also low, at 96.2–97.3% and 97.2–97.9% for VP2 and VP1, respectively. Blast analysis for both VP2 and VP1 nucleotide sequences of A3B2 Polish strains showed that the Israeli ks strain and the resulting attenuated mb strain were the most similar, with identities of 97.5–98.4% and 98.8–99.5%, respectively. It should be noted that A3B2 infections were detected only in the chicken flocks reared in eastern Poland, which may suggest a potential direction of virus introduction.

The last group A3c clustered the strains of genotype A3B4, which were earlier reported in Poland and Finland [28,29]. The nucleotide similarity of the A3.3 strains for VP2 was 97.1–100% and for VP1 97.9–100%. The phylogenetic analysis exhibited that these strains branched into two separate clades (Figure 1A). Interestingly, the first clustered the strains from IBD outbreaks from 2011 to 2018, while the strains from 2014 to 2022 formed the second clade, indicating that this emergent sublineage now dominates in the field.

The molecular survey carried out in one of the samples confirmed the presence of a strain belonging to serotype 2 (H286/17), which has not yet been reported in Poland. It showed the highest nucleotide similarity of the VP2 gene to the strain of European origin (23/82), which was 85.2%. In addition, one of the detected strains (97/1967/17) was on the same branch of the phylogenetic tree as the previously detected vvIBDV strains in Poland; nevertheless, its genotype could not be fully determined, as VP1 amplification failed.

During the study, the full-length genome sequencing of the 10 selected strains representing three genotypes was also performed. All the exhibited amino acid changes in the tested strains are shown in Figure 4 and Figure 5. The 38/22/Poland/2022 strain representing the A3B1 genotype possessed all aa changes previously reported by Mato et al. [19], including D^19^, L^74^, V^112^ in VP5; L^219^, D^254^, N^279^, and T^280^ in VP2 (Figure 4) and S^23^, L^275^, D^515^, and V^859^ in VP1 (Figure 5). However, the analysis confirmed the presence of additional eight aa substitutions, but four of them were also found in the Belgian A3B1 reassortant chicken/4439_001/2020 (Accession no. MZ367373 and MZ367374), such as R^133^ in VP5, K^359^ in VP2, N^787^ and V^951^ in VP3; and A^667^ in VP1 [30], which showed that there were at least two lineages within this group of reassortants. Unfortunately, only three full genomic sequences of A3B1 strains are available in the GenBank database, making it difficult to determine their diversity at the genetic level.

In turn, the Polish A3B2 strains share some molecular signatures with ks and mb strain, including N^279^, N^642^, S^745^, and V^990^ in the polyprotein encoded by segment A, as well as S^360^ within the VP1 protein [31]. However, additional amino acid changes were observed in the Polish A3B2 strains, such as K^221^ and D^254^ within the antigenic epitopes of VP2 protein (P_BC_, P_DE_, and P_FG_ loops, respectively), I^608^ in VP4, K^923^ in VP3, and D^246^ within VP1 (Figure 3 and Figure 4).

The strains of genotype A3B4 had the aa structure of polyprotein, which is typical for very virulent IBDV strains, except for the presence of threonine (T) at position 270 and valine (V) at position 990 (Figure 3). In turn, some of these strains had additional aa changes S^79^ in VP2 and I^263^ in VP1.

The phylogenetic analysis of the nucleotide sequences of the segments A (VP5-VP2-VP4-VP3) and B (VP1) also confirmed that the Polish strains belonged to the genotypes A3B1, A3B2, and A3B4 (Figure 6) and clustered into three different clades within genogroup 3.

### 3.2. Virus Isolation

The collected bursal samples (*n* = 40) were propagated using the chorioallantoic route on SPF embryonated eggs. All A3B4 (*n* = 22) strains were successfully isolated, and both embryo mortality between 3 and 6 dpi and macroscopic lesions, such as mottled-appearing necrosis in the liver and pale appearance of the heart, were noted during laboratory observation. Interestingly, during the isolation of the very virulent strains (*n* = 16), often neither mortality nor macroscopic changes were observed; moreover, in four cases, the virus could not be isolated. In turn, in the case of the two A3B1 reassortants, only strain 38/22/Poland/2022 was successfully isolated, and, as with the A3B2 strains, the virus did not cause mortality or lesions in the infected embryos.

## 4. Discussion

Poland is among the largest producer of chicken meat and eggs in the EU in recent years [32]. The high number and density of poultry flocks in some regions favor the spread of pathogens. On the other hand, extensive vaccination increases environmental pressure, which supports the selection of advantageous genetic changes and the emergence of new strains. IBDV infections pose a serious problem in poultry production worldwide due to direct or indirect losses. In Poland, the emergence of the vvIBDV was recognized at the end of 1991, the virus spread rapidly in the country leading to economic losses in both commercial and backyard farming [33]. A recent paper indicated the co-existence, for almost 30 years, of two IBDV lineages in the field, i.e., A3B2 and A3B4 genotypes [34]. Moreover, the A3B4 reassortants competed with the very virulent strains, becoming the prevalent lineage in Poland and disseminating to other countries. A similar situation is observed across Europe, hitherto dominated by vvIBDV, and now facing a wave of A3B1 IBDV infections [18,19]. Furthermore, the occurrence of a new A9B1 genotype was shown in Portugal [13] and antigenic variants in France [35]. The present study aimed to update the epidemiological situation of IBDV in Poland in order to detect potential threats to domestic poultry production.

A five-year molecular survey showed the presence of three genotypes of IBDV field strains, including the “old” A3B4 reassortants, and new A3B2 and A3B1, indicating the shift in the population of IBDV genetic lineages circulating in Poland. The results obtained established that the IBDVs of the A3B4 genotype have been continuously circulating in Poland since 1992, indicating the high fitness of this genetic lineage. The analysis showed further evolution of these strains (Figure 1 and Figure 5), highlighted by the bifurcation of the phylogenetic branch and the occurrence of additional aa alterations (Figure 3 and Figure 4). Most of the exhibited aa changes are individual; however, the presence of a serine at position 79 of VP2 and isoleucine at position 263 of VP1 was exhibited almost in all the strains, indicating the fixation of the previously resulting changes in subsequent generations of the virus progeny. Unfortunately, it is still unknown whether these changes affect the function of VP2 and VP1 proteins.

The next genotype of IBDV identified during the study was A3B2. Interestingly, these IBDVs represent newly introduced lineage, demonstrating that previously circulating lineage became extinct, or there was a strong decline in virus population size. The detected strain 97/1967/17 is likely to be the last identified representative of the previous vvIBDV lineage, as indicated by clustering together with their representatives, i.e., 75/11, 236/14, 72/15, and 20/15 (Figure 1A). The origin of this new A3B2 lineage in Poland is unknown. It should be added that there is no information on the occurrence of similar strains in Europe, and the absence of such strains in recent studies on numerous samples from western European countries suggests the opposite direction, especially given the availability of a very small number of such sequences in the GenBank database. This suspicion is also supported by the occurrence of these strains, which is limited to the eastern provinces of Poland (Figure 2B). Moreover, the phylogenetic analysis of the full-coding sequence revealed that the most closely related IBDV is the Israeli strain ks and the resulting attenuated mb strain. This mb strain is routinely used in Poland for immunoprophylaxis against IBD, which may suggest a potential origin of these strains. Nevertheless, this is contradicted by several data, the first being the presence of seven aa changes in the deduced sequence compared with the mb strain, including five in the polyprotein (Figure 3) and two in the viral polymerase (Figure 4). In addition, as a result of attenuation, two aa alterations were found in each of the two segments in the mb strain (T^272^ and T^527^; and N^96^ and A^161^ in segments A and B, respectively), whereas the Polish strain has only one (N^96^) from the listed changes [31]. Secondly, a vaccine containing the mb strain has been used in Poland since 2006, while the first field strains representing this genotype were found in 2017. It seems unlikely that the vaccine strain suddenly acquired such a high number of mutations after 10 years of use. Routine diagnostics carried out in our laboratory showed that the mb strain is one of the most stable vaccine strains, i.e., we did not record nucleotide changes in the hypervariable domain of the VP2 gene in the tested samples collected from vaccinated chickens. Another interesting aspect of this IBDV lineage is the presence of additional amino acid substitutions (K^221^, D^254^, and N^279^) located within P_BC_, P_DE_, and P_FG_ loops of VP2. One of the detected aa changes, i.e., N^279^ was previously attributed to IBDV adaptation to growth in cell culture [36] or the attenuation of IBDV [37]. During laboratory examination, the detected field strains were propagated using SPF embryonated eggs. Interestingly, these very virulent IBDVs proliferated relatively poorly; moreover, the virus did not cause any lesions in the infected embryos. However, only further research will provide an answer as to whether the changes detected altered the virulence or antigenicity of these strains.

The last IBDV detected during the survey represents the reassortants of the A3B1 genotype, which are now predominant in northwestern Europe [18,19]. The presence of A3B1 reassortants in Poland indicates the continued spread of these viruses in central Europe, as they have been confirmed in the Czech Republic and Slovakia [38]. The performed genetic analysis showed that the detected strains exhibited relatively low nucleotide similarity and possessed several aa alterations, indicating two independent virus introductions. The demonstrated diversity of A3B1 lineage reassortants is surprising, as these strains have circulated in Europe for 5 years [19], and the estimated tMRCA indicates the time of emergence of these strains between 2015 and 2017 [18]. What is also surprising is the scale of the extent of this epidemic. The spread of these lineages in Europe is probably favored by the reduced virulence [19], resulting in a subclinical course of infection, which makes the proper diagnosis of the disease difficult. Moreover, these strains have a number of aa alterations in antigenic epitopes, which may favor evasion of the post-vaccination response. Previous studies reported that one of the observed aa changes (D^254^) resulted in altered reactivity with monoclonal antibodies (Mab5) of A4B1 genotype strains [39]. However, only studies of the efficacy of IBD immunoprophylaxis will confirm this theory.

In conclusion, the results obtained exhibited a shift in the genetic lineages of IBDV circulating in Poland. In addition, three different IBDV genotypes were demonstrated in the field, indicating a more complex epidemiological situation than before. The introduction of two genotypes carrying a number of aa substitutions within the antigenic determinants was found, raising the question of whether the vaccines used confer protection against their infection. In addition, we demonstrated the further spread of A3B1 reassortants in central Europe, showing the need for further monitoring studies in this region and an assessment of their impact on poultry welfare in the EU.

## Figures and Tables

**Figure 1 viruses-15-00289-f001:**
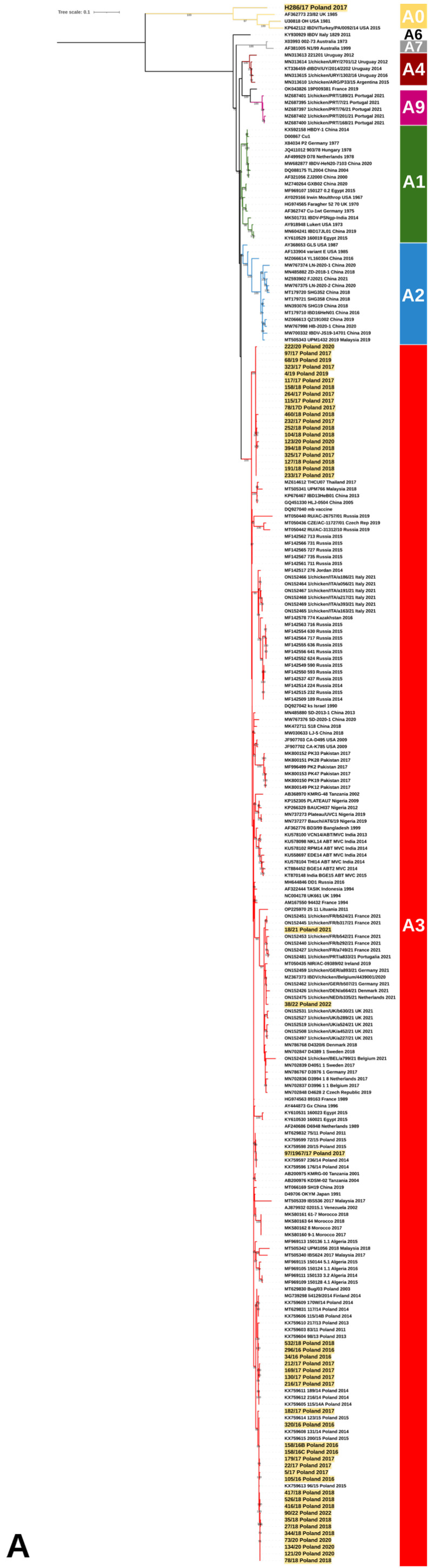
Maximum likelihood phylogenetic analysis of the truncated sequence of (**A**) VP2 (561nt) and (**B**) VP1 (427nt) genes according to Wang et al. rules [11]. Trees were constructed using GTR + F + I + G4 and TIM2e + G4 with 1000 bootstrap iterations for VP2 and VP1, respectively. The main IBDV genogroups are denoted with designation and colors, and the taxa name of the Polish strains characterized in the presented study are highlighted with yellow.

**Figure 2 viruses-15-00289-f002:**
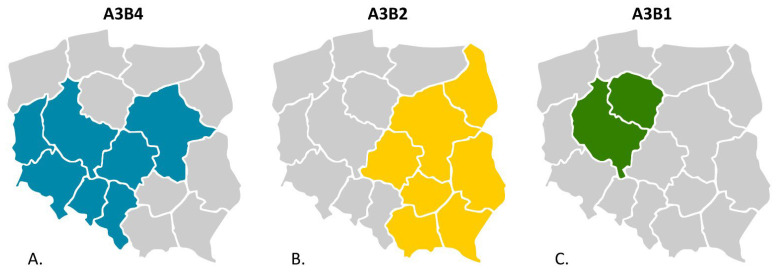
Depiction of administrative areas in Poland where IBDV field strains belonging to A3B4 (**A**), A3B2 (**B**), and A3B1 (**C**) genotypes were detected.

**Figure 3 viruses-15-00289-f003:**
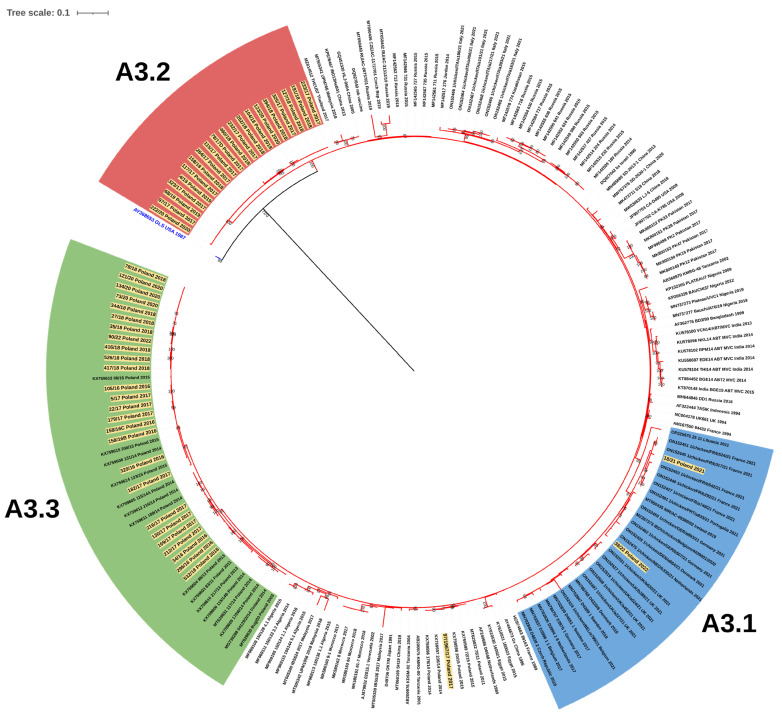
Subtree for the genogroup A3, the clades for the Polish strains were denoted A3.1 in blue, A3.2 in red, and A3.3 in green, and the sequence AY368653 was outgroup (blue).

**Figure 4 viruses-15-00289-f004:**
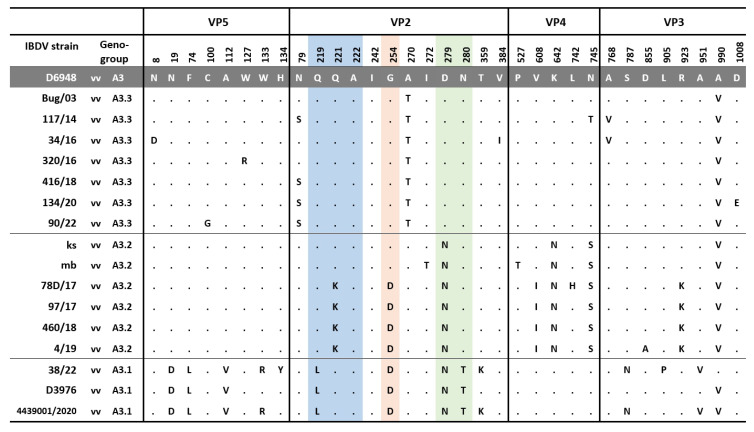
Comparison of amino acid substitutions in viral proteins encoded by segment A between Polish and reference strains. Dots show identity with D6948. The amino acids forming the P_BC_ (aa 219–224), P_DE_ (aa 249–254), and P_FG_ (aa 279–284) loops of the VP2 hypervariable domain are highlighted in blue, orange, and green, respectively. Abbreviations: vv—very virulent, A—alanine, D—aspartic acid, E—glutamic acid, F—phenylalanine, G—glycine, H—histidine, I –isoleucine, K—lysine, L—leucine, N—asparagine, S—serine, R—arginine, T—threonine, V—valine, Y—tyrosine.

**Figure 5 viruses-15-00289-f005:**
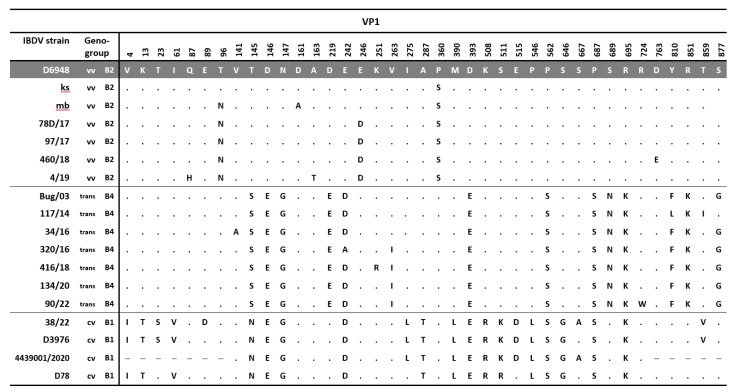
Comparison of amino acid substitutions in viral proteins encoded by segment B between Polish and reference strains. Dots show identity with D6948, while dashes indicate the lack of available aa sequence. Abbreviations: vv—very virulent, trans—transitional IBDV, cv—classical virulent, A—alanine, D—aspartic acid, E—glutamic acid, F—phenylalanine, G—glycine, H-histidine, I—isoleucine, K—lysine, L-leucine, N—asparagine, S—serine, R—arginine, T—threonine, V—valine, W—tryptophan.

**Figure 6 viruses-15-00289-f006:**
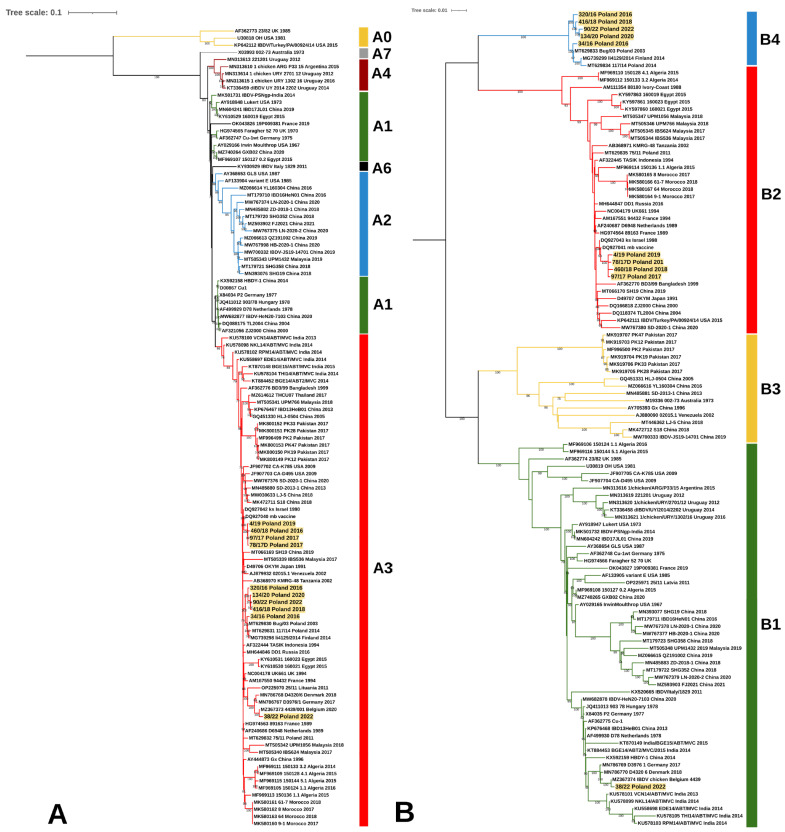
Maximum likelihood phylogenetic analysis of full coding sequence of (**A**) segment A (3076 nt) and (**B**) B (2640 nt). Trees were constructed using GTR + F + I + G4 with 1000 bootstrap iterations. The main IBDV genogroups are denoted with designation and colors, and the taxa name of the Polish strains characterized in the presented study are highlighted with yellow.

**Table 1 viruses-15-00289-t001:** Primers used for amplification of IBDV in the study.

Primers	Positions ^1^	Nucleotide Sequence (5′→3′)	Amplicon Size (bp)
Segment A			
VP2-F	512	ACCTTCCAAGGAAGCCTGAGTG	739
VP2-R	1251	ATCAGCTCGAAGTTGCTCACC	
SegA0_F	1	GGATACGATCGGTCTGACCCCGGGGGAG	3260
SegA0_R	3260	GGGGACCCGCGAACGGATCCAAT	
Segment B			
VP1.154F	154	TCAGCAGCGTTCGGCATAAAGC	790
VP1.998R	998	TTTGTCCCTGCACCCTGCTTCA	
SegB0_F	1	GGATACGATGGGTCTGACCCTCTG	2827
SegB0_R	2827	GGGGGCCCCCGCAGGCGAAG	

^1^ Numbering according to D6948 strain (segment A: AF240686, segment B: AF240687).

**Table 2 viruses-15-00289-t002:** List of field strains circulated in Poland 2016–2022.

Strain	Genotype	Collection	Chicken		Vaccination	GenBank Accession no.	
		Date	Age	Type	(Vaccine Strain)	VP1	VP2
34/16	A3B4	February 2016	5w	broiler	LIBDV	**OP978030**	**OP978040**
105/16	A3B4	April 2016	42d	broiler	Lukert	OP978050	OP978090
158/16B	A3B4	June 2016	6w	broiler	NA	OP978051	OP978091
158/16C	A3B4	June 2016	4w	broiler	NA	OP978052	OP978092
296/16	A3B4	November 2016	31d	broiler	W2512 & CH/80	OP978053	OP978093
320/16	A3B4	December 2016	6w	broiler	D78	**OP978031**	**OP978041**
5/17	A3B4	January 2017	4w	broiler	LC75	OP978054	OP978094
22/17	A3B4	January 2017	18w	layer	LIBDV	OP978055	OP978095
H287/17	A0Bx	March 2017	NA	broiler	NA	OP978056	-
78/17	A3B2	March 2017	43d	broiler	LC75	**OP978032**	**OP978042**
97/17	A3B2	April 2017	NA	broiler	D78	**OP978033**	**OP978043**
97/1967/17	A3Bx	April 2017	NA	broiler	NA	OP978057	-
115/17	A3B2	April 2017	5.5w	broiler	GM97	OP978058	OP978096
117/17	A3B2	April 2017	NA	broiler	W2512 & D78	OP978059	OP978097
130/17K10	A3B4	April 2017	31d	broiler	W2512 & CH/80	OP978060	OP978098
130/17K11	A3B4	April 2017	31d	broiler	W2512 & CH/80	OP978061	OP978098
169/17	A3B4	May 2017	25d	broiler	NA	OP978061	OP978099
179/17	A3B4	June 2017	9w	layer	NA	OP978062	OP978100
182/17	A3B4	June 2017	6w	broiler	Faragher 52/70	OP978063	OP978101
212/17	A3B4	June 2017	NA	NA	NA	OP978064	OP978102
216/17	A3B4	June 2017	NA	NA	GM97	OP978065	OP978103
232/17	A3B2	July 2017	6w	broiler	GM97	OP978066	OP978104
233/17	A3B2	July 2017	6w	broiler	GM97	OP978067	OP978105
264/17	A3B2	August 2017	39d	broiler	GM97 & CH/80	OP978068	OP978106
323/17	A3B2	October 2017	41d	broiler	LIBDV	OP978069	OP978107
325/17	A3B2	October 2017	39d	broiler	GM97	OP978070	OP978108
27/18	A3B4	February 2018	31d	broiler	NA	OP978071	OP978109
35/18	A3B4	February 2018	5w	broiler	NA	OP978072	OP978110
78/18	A3B4	March 2018	5w	broiler	NA	OP978073	OP978111
104/18K5	A3B2	April 2018	5.5w	broiler	NA	OP978074	OP978112
104/18K7	A3B2	April 2018	5w	broiler	NA	OP978074	OP978112
127/18	A3B2	April 2018	49d	broiler	NA	OP978075	OP978113
158/18	A3B2	May 2018	40d	layer	No vac^1^	OP978076	OP978114
191/18	A3B2	June 2018	32d	broiler	GM97	OP978077	OP978115
252/18	A3B2	July 2018	NA	broiler	NA	OP978078	OP978116
344/18	A3B4	July 2018	6w	layer	Faragher 52/70	OP978079	OP978117
394/18	A3B2	August 2018	32d	broiler	NA	OP97808	OP978118
416/18	A3B4	September 2018	36d	broiler	W2512 & LIBDV	**OP978034**	**OP978044**
417/18	A3B4	September 2018	32d	broiler	W2512 & LIBDV	OP978081	OP978119
460/18	A3B2	October 2018	35d	broiler	W2512	**OP978035**	**OP978045**
526/18	A3B4	November 2018	7w	layer	CH/80	OP978082	OP978120
532/18	A3B4	November 2018	4.5w	broiler	W2512	OP978083	OP978121
4/19	A3B2	January 2019	5w	broiler	NA	**OP978036**	**OP978046**
68/19	A3Bx	February 2019	NA	broiler	GM97	OP978084	-
73/20	A3B4	April 2020	6w	layer	D78	OP978085	OP978122
121/20	A3B4	May 2020	39d	broiler	No vac^1^	OP978086	OP978123
123/20	A3B2	May 2020	6w	broiler	NA	OP97808	OP978124
134/20	A3B4	June 2020	NA	broiler	NA	**OP978037**	**OP978047**
222/20	A3B2	August 2020	35d	broiler	NA	OP978088	OP978125
18/21	A3B1	August 2021	42d	broiler	NA	OP978089	OP978126
38/22	A3B1	March 2022	31d	broiler	LIBDV	**OP978038**	**OP978048**
90/22	A3B4	June 2022	4w	layer	1/65/PV	**OP978039**	**OP978049**

d: days; w: weeks; NA: not available; No vac—no vaccination; ^1^ organic production; full-coding sequences are denoted in bold.

## Data Availability

The obtained genome sequences generated in this study were submitted to the GenBank database (https://www.ncbi.nlm.nih.gov/genbank/ accessed on 24 November 2022, under accession numbers OP978030 and OP978126).

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
