# Peer review of "Epidemiology of Infectious Bursal Disease Virus in Poland during 2016–2022"

_viruses, 2023, doi:10.3390/v15020289_

Round 1
Reviewer 1 Report
Dear authors
Minor editing is suggested, as follows
L2-3 The title could be improved. "Epidemiologic surveillance study of infectious bursal disease virus from 2016 to 2022 in Poland" could be an option to consider.
L31 Please consider IBDV is not the only member of the Avibirnavirus genus and Birnavirividae family. Currently, the chicken proventricular necrosis virus causing transmissible viral proventriculitis is considered a new member of the mentioned genus and family.
L64-66 Please include the criteria for the bursal tissue sampling included in this study. Do you consider associated clinical signs, gross findings, or serologic studies with the bursal sampling? Can you add additional information about the studied birds such as age, location, and IBD vaccine status?
L89 Please change from "The obtained all PCR products..." to "All obtained PCR products..."
L106 Please use "specific pathogen free (SPF)"
L123 Please clarify if the named 20 chicken flocks belonged to broiler and/or laying chickens.
L159 Please clarify this sentence when you refer to "ks strain" and mb strain"
L163 Please delete ", which".
L 209, 250, 274, 298, 305, 312 Please use "aa" instead of "amino acids"
L228-230 Please insert reference/references to support this paragraph.
Author Response
Dear reviewer,
Thank you very much for the revision and all suggestions. Below is a detailed response to the comments.
1) L2-3 The title could be improved. "Epidemiologic surveillance study of infectious bursal disease virus from 2016 to 2022 in Poland" could be an option to consider.
Response to comment: Thank you very much for the suggestion, however, both titles are similar, and the indicated change is the subjective perception of the reviewer.
2) L31 Please consider IBDV is not the only member of the Avibirnavirus genus and Birnavirividae family. Currently, the chicken proventricular necrosis virus causing transmissible viral proventriculitis is considered a new member of the mentioned genus and family.
Response to comment: Chicken proventricular necrosis virus is an unclassified virus of Birnaviridea family therefore we indicated that IBDV is the only representative of the Avibirnavir genus. However, given that it also infects birds we have revised the sentence and remove “the only”.
3) L64-66 Please include the criteria for the bursal tissue sampling included in this study. Do you consider associated clinical signs, gross findings, or serologic studies with the bursal sampling? Can you add additional information about the studied birds such as age, location, and IBD vaccine status?
Response to comment: All samples were sent to NVRI for diagnostic examination and were taken from flocks suspected of be infected with IBDV, MDV or adenovirus. Relevant information was added in lines 66-67. Some additional information such as age and vaccination are presented in Table 2. While, in order to show the location of IBDV strains of each genotype, Figure 2 was introduced.
4) L89 Please change from "The obtained all PCR products..." to "All obtained PCR products..."
Response to comment: amendments were made as suggested by reviewer
5) L106 Please use "specific pathogen free (SPF)"
Response to comment: amendments were made as suggested by reviewer
6) L123 Please clarify if the named 20 chicken flocks belonged to broiler and/or laying chickens.
Response to comment: Details of the positive flocks are shown in Table 2, including the type of chicken breed.
7) L159 Please clarify this sentence when you refer to "ks strain" and mb strain" Response to comment: amendments were made see line 164.
8) L163 Please delete ", which". Response to comment: amendments were made as suggested by reviewer
9) L 209, 250, 274, 298, 305, 312 Please use "aa" instead of "amino acids" Response to comment: amendments were made as suggested by reviewer
10) L228-230 Please insert reference/references to support this paragraph.
Response to comment: As a reference, we have added Eurostat data available at https://ec.europa.eu/eurostat/databrowser/view/tag00043/default/table?lang=en
Reviewer 2 Report
Please refer to the attached report.

Author Response
Dear reviewer,
Thank you very much for the revision and all suggestions. Below is a detailed response to the comments.
- The VP2 of field strains were found to fall into three separate clusters within genogroup A3, which were named A3a, A3b and A3c. However, since their boundaries are not based on objective thresholds and their usefulness is limited to this study (to describe the different types of field strains), I would be wary of naming then in such a way. The same approach (genogroup name + another letter) is being used by different research groups (for the Chinese variant IBDVs, for instance) based on different classification criteria, and I fear that on the long run this could generate some confusion. For these reasons, although I understand the reasoning behind the distinction, I would suggest using an alternative nomenclature (more descriptive, perhaps?) for the three clusters if possible.
Response to comment: Thank you for the remark, in order to avoid misclassification we have corrected the clade names to A3.1, A3.2 and A3.3. - Line 114. The percentage of field strain detections over the number of investigated flocks could be provided on a yearly basis, too.
Response to comment: Thank you for the suggestions, an appropriate amendments were made into the manuscript, see lines 115-118. - Table 2. The column in which the productive type is indicated is wrongly titled “date”.
Response to comment: Thank you for pointing out the error, which probably occurred during the transfer of data. Appropriate amendments were made into manuscript. - Table 2. Every vaccine but Vaxxitek is listed by giving the vaccine strain rather than the commercial name. To stay consistent, I would replace Vaxxitek with “Faragher 52/70”, since it expresses the VP2 of that strain.
Response to comment: Thank you for suggestions, an appropriate amendments were made into Table 2. - Although the manuscript is understandable from a language standpoint, there are some errors throughout the text. Below are some suggestions, but I would make sure to thoroughly revise the manuscript for other possible inaccuracies:
o Line 8 -> “losses” instead of “loses”
o Line 9 -> “drive” instead of “drives”; “lead” instead of “leads”
o Line 25 -> “is a ubiquitous pathogen that leads to”
o Line 26 -> “replicates” instead of “replicate”
o Line 34 -> “cleaved” instead of “cleavage”
o Line 38 -> “contributes to”
o Line 41 -> “continuous” instead of “continues”
o Line 96 -> Delete “of” after “align”
o Line 121 -> “were” instead of “was”
o Line 152 -> I believe “introductions” would be preferable instead of “introduction”
o Line 230 -> “Pathogens”,“infectious agents”, etc. instead of “pathogen infections”
o Line 261 -> “these IBDVs represent a newly introduced lineage”
Response to comment: All the errors mentioned above were corrected according to suggestions